# Noise-Enhanced Associative Memories

**Amin Karbasi**
Swiss Federal Institute of Technology Zurich
`amin.karbasi@inf.ethz.ch`

**Amir Hesam Salavati**
Ecole Polytechnique Federale de Lausanne
`hesam.salavati@epfl.ch`

**Amin Shokrollahi**
Ecole Polytechnique Federale de Lausanne
`amin.shokrollahi@epfl.ch`

**Lav R. Varshney**
IBM Thomas J. Watson Research Center
`varshney@alum.mit.edu`

## Abstract

Recent advances in associative memory design through structured pattern sets and graph-based inference algorithms allow reliable learning and recall of exponential numbers of patterns. Though these designs correct external errors in recall, they assume neurons compute noiselessly, in contrast to highly variable neurons in hippocampus and olfactory cortex. Here we consider associative memories with noisy internal computations and analytically characterize performance. As long as internal noise is less than a specified threshold, error probability in the recall phase can be made exceedingly small. More surprisingly, we show internal noise actually improves performance of the recall phase. Computational experiments lend additional support to our theoretical analysis. This work suggests a functional benefit to noisy neurons in biological neuronal networks.

## 1   Introduction

Hippocampus, olfactory cortex, and other brain regions are thought to operate as associative memories [1,2], having the ability to learn patterns from presented inputs, store a large number of patterns, and retrieve them reliably in the face of noisy or corrupted queries [3–5]. Associative memory models are designed to have these properties.

Although such information storage and recall seemingly falls into the information-theoretic framework, where an exponential number of messages can be communicated reliably with a linear number of symbols, classical associative memory models could only store a linear number of patterns [4]. A primary reason is classical models require memorizing a randomly chosen set of patterns. By enforcing structure and redundancy in the possible set of memorizable patterns—like natural stimuli [6], internal neural representations [7], and error-control codewords—advances in associative memory design allow storage of an exponential number of patterns [8,9], just like in communication systems.

Information-theoretic and associative memory models of storage have been used to predict experimentally measurable properties of synapses in the mammalian brain [10,11]. But contrary to the fact that noise is present in computational operations of the brain [12, 13], associative memory models with exponential capacities have assumed no internal noise in the computational nodes. The purpose here is to model internal noise and study whether such associative memories still operate reliably. Surprisingly, we find internal noise actually enhances recall performance, suggesting a functional role for variability in the brain.

In particular we consider a multi-level, graph code-based, associative memory model [9] and find that even if all components are noisy, the final error probability in recall can be made exceedingly small. We characterize a threshold phenomenon and show how to optimize algorithm parameters when knowing statistical properties of internal noise. Rather counterintuitively the performance

of the memory model *improves* in the presence of internal neural noise, as observed previously as *stochastic resonance* [13, 14]. There are mathematical connections to perturbed simplex algorithms for linear programing [15], where internal noise pushes the algorithm out of local minima.

The benefit of internal noise has been noted previously in associative memory models with stochastic update rules, cf. [16]. However, our framework differs from previous approaches in three key aspects. First, our memory model is different, which makes extension of previous analysis nontrivial. Second, and perhaps most importantly, pattern retrieval capacity in previous approaches *decreases* with internal noise, cf. [16, Fig. 6.1], in that increasing internal noise helps correct more external errors, but also reduces the number of memorizable patterns. In our framework, internal noise does not affect pattern retrieval capacity (up to a threshold) but improves recall performance. Finally, our noise model has bounded rather than Gaussian noise, and so a suitable network may achieve *perfect* recall despite internal noise.

Reliably storing information in memory systems constructed completely from unreliable components is a classical problem in fault-tolerant computing [17–19], where models have used random access architectures with sequential correcting networks. Although direct comparison is difficult since notions of circuit complexity are different, our work also demonstrates that associative memory architectures constructed from unreliable components can store information reliably.

Building on the idea of structured pattern sets [20], our associative memory model [9] relies on the fact that all patterns to be learned lie in a low-dimensional subspace. Learning features of a low-dimensional space is very similar to autoencoders [21] and has structural similarities to Deep Belief Networks (DBNs), particularly Convolutional Neural Networks [22].

## 2 Associative Memory Model

**Notation and basic structure**: In our model, a neuron can assume an integer-valued state from the set $\mathcal{S} = \{0, \ldots, S-1\}$, interpreted as the short term firing rate of neurons. A neuron updates its state based on the states of its neighbor $\{s_i\}_{i=1}^n$ as follows. It first computes a weighted sum $h = \sum_{i=1}^n w_i s_i + \zeta$, where $w_i$ is the weight of the link from $s_i$ and $\zeta$ is the *internal noise*, and then applies nonlinear function $f : \mathbb{R} \to \mathcal{S}$ to $h$.

An associative memory is represented by a weighted bipartite graph, $G$, with pattern neurons and constraint neurons. Each pattern $x = (x_1, \ldots, x_n)$ is a vector of length $n$, where $x_i \in \mathcal{S}, i = 1, \ldots, n$. Following [9], the focus is on recalling patterns with strong *local correlation* among entries. Hence, we divide entries of each pattern $x$ into $L$ *overlapping* sub-patterns of lengths $n_1, \ldots, n_L$. Due to overlaps, a pattern neuron can be a member of multiple subpatterns, as in Fig. 1a. The $i$th subpattern is denoted $x^{(i)} = (x_1^{(i)}, \ldots, x_{n_i}^{(i)})$, and local correlations are assumed to be in the form of subspaces, i.e. the subpatterns $x^{(i)}$ form a subspace of dimension $k_i < n_i$.

We capture the local correlations by learning a set of linear constraints over each subspace corresponding to the dual vectors orthogonal to that subspace. More specifically, let $\{w_1^{(i)}, \ldots, w_{m_i}^{(i)}\}$ be a set of dual vectors orthogonal to all subpatterns $x^{(i)}$ of cluster $i$. Then:

$$y_j^{(i)} = (w_j^{(i)})^T \cdot x^{(i)} = 0, \quad \text{for all } j \in \{1, \ldots, m_i\} \text{ and for all } i \in \{1, \ldots, L\}. \tag{1}$$

Eq. (1) can be rewritten as $W^{(i)} \cdot x^{(i)} = 0$ where $W^{(i)} = [w_1^{(i)} | w_2^{(i)} | \ldots | w_{m_i}^{(i)}]^T$ is the matrix of dual vectors. Now we use a bipartite graph with connectivity matrix determined by $W^{(i)}$ to represent the subspace constraints learned from subpattern $x^{(i)}$; this graph is called *cluster $i$*. We developed an efficient way of learning $W^{(i)}$ in [9], also used here. Briefly, in each iteration of learning:

1. Pick a pattern $x$ at random from the dataset;

2. Adjust weight vectors $w_j^{(i)}$ for $j = \{1, \ldots, m_i\}$ and $i = \{1, \ldots, L\}$ such that the projection of $x$ onto $w_j^{(i)}$ is reduced. Apply a sparsity penalty to favor sparse solutions.

This process repeats until all weights are orthogonal to the patterns in the dataset or the maximum iteration limit is reached. The learning rule allows us to assume the weight matrices $W^{(i)}$ are known and satisfy $W^{(i)} \cdot x^{(i)} = 0$ for all patterns $x$ in the dataset $\mathcal{X}$, in this paper.

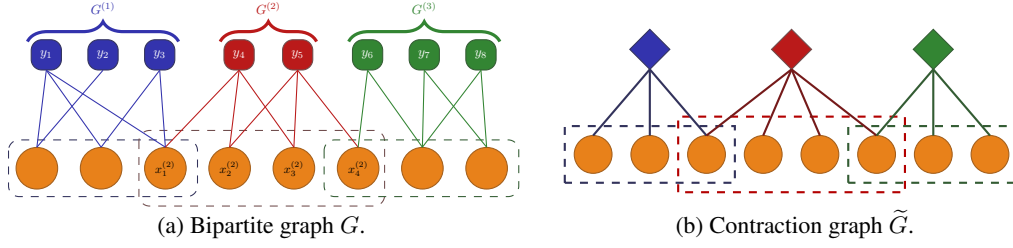

(a) Bipartite graph $G$.

(b) Contraction graph $\widetilde{G}$.

Figure 1: The proposed neural associative memory with overlapping clusters.

For the forthcoming asymptotic analysis, we need to define a *contracted graph* $\widetilde{G}$ whose connectivity matrix is denoted $\widetilde{W}$ and has size $L \times n$. This is a bipartite graph in which constraints in each cluster are represented by a single neuron. Thus, if pattern neuron $x_j$ is connected to cluster $i$, $\widetilde{W}_{ij} = 1$; otherwise $\widetilde{W}_{ij} = 0$. We also define the degree distribution from an *edge perspective* over $\widetilde{G}$, using $\widetilde{\lambda}(z) = \sum_j \widetilde{\lambda}_j z^j$ and $\widetilde{\rho}(z) = \sum_j \widetilde{\rho}_j z^{j-1}$ where $\widetilde{\lambda}_j$ (resp., $\widetilde{\rho}_j$) equals the fraction of edges that connect to pattern (resp., cluster) nodes of degree $j$.

**Noise model**: There are two types of noise in our model: *external errors* and *internal noise*. As mentioned earlier, a neural network should be able to retrieve memorized pattern $\hat{x}$ from its corrupted version $x$ due to external errors. We assume the external error is an additive vector of size $n$, denoted by $z$ satisfying $x = \hat{x} + z$, whose entries assume values independently from $\{-1, 0, +1\}$[1] with corresponding probabilities $p_{-1} = p_{+1} = \epsilon/2$ and $p_0 = 1 - \epsilon$. The realization of the external error on subpattern $x^{(i)}$ is denoted $z^{(i)}$. Note that the subspace assumption implies $W \cdot y = W \cdot z$ and $W^{(i)} \cdot y^{(i)} = W^{(i)} \cdot z^{(i)}$ for all $i$. Neurons also suffer from internal noise. We consider a bounded noise model, i.e. a random number uniformly distributed in the intervals $[-\upsilon, \upsilon]$ and $[-\nu, \nu]$ for the pattern and constraint neurons, respectively ($\upsilon, \nu < 1$).

The goal of recall is to filter the external error $z$ to obtain the desired pattern $x$ as the correct states of the pattern neurons. When neurons compute noiselessly, this task may be achieved by exploiting the fact the set of patterns $x \in \mathcal{X}$ to satisfy the set of constraints $W^{(i)} \cdot x^{(i)} = 0$. However, it is not clear how to accomplish this objective when the neural computations are noisy. Rather surprisingly, we show that eliminating external errors is not only possible in the presence of internal noise, but that neural networks with moderate internal noise demonstrate better external noise resilience.

**Recall algorithms**: To efficiently deal with external errors, we use a combination of Alg. 1 and Alg. 2. The role of Alg. 1 is to correct at least a single external error in each cluster. Without overlaps between clusters, the error resilience of the network is limited. Alg. 2 exploits the overlaps: it helps clusters with external errors recover their correct states by using the reliable information from clusters that do not have external errors. The error resilience of the resulting combination thereby drastically improves. Now we describe the details of Alg. 1 and Alg. 2 more precisely.

Alg. 1 performs a series of forward and backward iterations in each cluster $G^{(l)}$ to remove (at least) one external error from its input domain. At each iteration, the pattern neurons locally decide whether to update their current state: if the amount of feedback received by a pattern neuron exceeds a threshold, the neuron updates its state, and otherwise remains as is. With abuse of notation, let us denote messages transmitted by pattern node $i$ and constraint node $j$ at round $t$ by $x_i(t)$ and $y_j(t)$, respectively. In round 0, pattern nodes are initialized by a pattern $\hat{x}$, sampled from dataset $\mathcal{X}$, perturbed by external errors $z$, i.e. $x(0) = \hat{x} + z$. Thus, for cluster $\ell$ we have $x^{(\ell)}(0) = \hat{x}^{(\ell)} + z^{(\ell)}$, where $z^{(\ell)}$ is the realization of errors on subpattern $x^{(\ell)}$.

In round $t$, the pattern and constraint neurons update their states using feedback from neighbors. However since neural computations are faulty, decisions made by neurons may not be reliable. To minimize effects of internal noise, we use the following update rule for pattern node $i$ in cluster $\ell$:

$$x_i^{(\ell)}(t+1) = \begin{cases} x_i^{(\ell)}(t) - \text{sign}(g_i^{(\ell)}(t)), & \text{if } |g_i^{(\ell)}(t)| \geq \varphi \\ x_i^{(\ell)}(t), & \text{otherwise,} \end{cases} \qquad (2)$$

| **Algorithm 1** Intra-Module Error Correction | **Algorithm 2** Sequential Peeling Algorithm |
|---|---|
| **Input:** Training set $\mathcal{X}$, thresholds $\varphi, \psi$, iteration $t_{\max}$<br>**Output:** $x_1^{(\ell)}, x_2^{(\ell)}, \ldots, x_{n_\ell}^{(\ell)}$<br> 1: **for** $t = 1 \to t_{\max}$ **do**<br> 2:   *Forward iteration:* Calculate the input $h_i^{(\ell)} = \sum_{j=1}^{n_\ell} W_{ij}^{(\ell)} x_j^{(\ell)} + v_i$, for each neuron $y_i^{(\ell)}$ and set $y_i^{(\ell)} = f(h_i^{(\ell)}, \psi)$.<br> 3:   *Backward iteration:* Each neuron $x_j^{(\ell)}$ computes<br>     $g_j^{(\ell)} = \frac{\sum_{i=1}^{m_\ell} \mathrm{sign}(W_{ij}^{(\ell)}) y_i^{(\ell)}}{\sum_{i=1}^{m_\ell} \mathrm{sign}(|W_{ij}^{(\ell)}|)} + u_i.$<br> 4:   Update state of each pattern neuron $j$ according to $x_j^{(\ell)} = x_j^{(\ell)} - \mathrm{sign}(g_j^{(\ell)})$ only if $|g_j^{(\ell)}| > \varphi$.<br> 5: **end for** | **Input:** $\widetilde{G}, G^{(1)}, G^{(2)}, \ldots, G^{(L)}$.<br>**Output:** $x_1, x_2, \ldots, x_n$<br> 1: **while** there is an unsatisfied $v^{(\ell)}$ **do**<br> 2:   **for** $\ell = 1 \to L$ **do**<br> 3:     If $v^{(\ell)}$ is unsatisfied, apply Alg. 1 to cluster $G^{(l)}$.<br> 4:     If $v^{(\ell)}$ remained unsatisfied, revert state of pattern neurons connected to $v^{(\ell)}$ to their initial state. Otherwise, keep their current states.<br> 5:   **end for**<br> 6: **end while**<br> 7: Declare $x_1, x_2, \ldots, x_n$ if all $v^{(\ell)}$'s are satisfied. Otherwise, declare failure. |

where $\varphi$ is the update threshold and $g_i^{(\ell)}(t) = \left( (\mathrm{sign}(W^{(\ell)})^\top \cdot y^{(\ell)}(t))_i / d_i^{(\ell)} + u_i. \right.$[2] Here, $d_i^{(\ell)}$ is the degree of pattern node $i$ in cluster $\ell$, $y^{(\ell)}(t) = [y_1^{(\ell)}(t), \ldots, y_{m_\ell}^{(\ell)}(t)]$ is the vector of messages transmitted by the constraint neurons in cluster $\ell$, and $u_i$ is the random noise affecting pattern node $i$. Basically, the term $g_i^{(\ell)}(t)$ reflects the (average) belief of constraint nodes connected to pattern neuron $i$ about its correct value. If $g_i^{(\ell)}(t)$ is larger than a specified threshold $\varphi$ it means most of the connected constraints suggest the current state $x_i^{(\ell)}(t)$ is not correct, hence, a change should be made. Note this average belief is diluted by the internal noise of neuron $i$. As mentioned earlier, $u_i$ is uniformly distributed in the interval $[-\upsilon, \upsilon]$, for some $\upsilon < 1$. On the constraint side, the update rule is:

$$y_i^{(\ell)}(t) = f(h_i^{(\ell)}(t), \psi) = \begin{cases} +1, & \text{if } h_i^{(\ell)}(t) \geq \psi \\ 0, & \text{if } -\psi \leq h_i^{(\ell)}(t) \leq \psi \\ -1, & \text{otherwise,} \end{cases} \qquad (3)$$

where $\psi$ is the update threshold and $h_i^{(\ell)}(t) = \left( W^{(\ell)} \cdot x^{(\ell)}(t) \right)_i + v_i$. Here, $x^{(\ell)}(t) = [x_1^{(\ell)}(t), \ldots, x_{n_\ell}^{(\ell)}(t)]$ is the vector of messages transmitted by the pattern neurons and $v_i$ is the random noise affecting node $i$. As before, we consider a bounded noise model for $v_i$, i.e., it is uniformly distributed in the interval $[-\nu, \nu]$ for some $\nu < 1$.[3]

The error correction ability of Alg. 1 is fairly limited, as determined analytically and through simulations [23]. In essence, Alg. 1 can correct one external error with high probability, but degrades terribly against two or more external errors. Working independently, clusters cannot correct more than a few external errors, but their combined performance is much better. As clusters overlap, they help each other in resolving external errors: a cluster whose pattern neurons are in their correct states can *always* provide truthful information to neighboring clusters. This property is exploited in Alg. 2 by applying Alg. 1 in a round-robin fashion to each cluster. Clusters either eliminate their internal noise in which case they keep their new states and can now help other clusters, or revert back to their original states. Note that by such a scheduling scheme, neurons can only change their states towards correct values. This scheduling technique is similar in spirit to the peeling algorithm [24].

## 3 Recall Performance Analysis

Now let us analyze recall error performance. The following lemma shows that if $\varphi$ and $\psi$ are chosen properly, then in the absence of external errors the constraints remain satisfied and internal noise cannot result in violations. This is a crucial property for Alg. 2, as it allows one to determine whether

a cluster has successfully eliminated external errors (Step 4 of algorithm) by merely checking the satisfaction of all constraint nodes.

**Lemma 1.** *In the absence of external errors, the probability that a constraint neuron (resp. pattern neuron) in cluster $\ell$ makes a wrong decision due to its internal noise is given by $\pi_0^{(\ell)} = \max\left(0, \frac{\nu - \psi}{\nu}\right)$ (resp. $P_0^{(\ell)} = \max\left(0, \frac{\upsilon - \varphi}{\upsilon}\right)$).*

Proof is given in [23]. In the sequel, we assume $\varphi > \upsilon$ and $\psi > \nu$ so that $\pi_0^{(\ell)} = 0$ and $P_0^{(\ell)} = 0$. However, an external error combined with internal noise may still push neurons to an incorrect state.

Given the above lemma and our neural architecture, we can prove the following surprising result: in the asymptotic regime of increasing number of iterations of Alg. 2, a neural network with internal noise outperforms one without. Let us define the fraction of errors corrected by the noiseless and noisy neural network (parametrized by $\upsilon$ and $\nu$) after $T$ iterations of Alg. 2 by $\Lambda(T)$ and $\Lambda_{\upsilon,\nu}(T)$, respectively. Note that both $\Lambda(T) \leq 1$ and $\Lambda_{\upsilon,\nu}(T) \leq 1$ are non-decreasing sequences of $T$. Hence, their limiting values are well defined: $\lim_{T\to\infty} \Lambda(T) = \Lambda^*$ and $\lim_{T\to\infty} \Lambda_{\upsilon,\nu}(T) = \Lambda_{\upsilon,\nu}^*$.

**Theorem 2.** *Let us choose $\varphi$ and $\psi$ so that $\pi_0^{(\ell)} = 0$ and $P_0^{(\ell)} = 0$ for all $\ell \in \{1, \ldots, L\}$. For the same realization of external errors, we have $\Lambda_{\upsilon,\nu}^* \geq \Lambda^*$.*

Proof is given in [23]. The high level idea why a noisy network outperforms a noiseless one comes from understanding stopping sets. These are realizations of external errors where the iterative Alg. 2 cannot correct all of them. We show that the stopping set shrinks as we add internal noise. In other words, we show that in the limit of $T \to \infty$ the noisy network can correct any error pattern that can be corrected by the noiseless version and it can also get out of stopping sets that cause the noiseless network to fail. Thus, the supposedly harmful internal noise will help Alg. 2 to avoid stopping sets.

Thm. 2 suggests the only possible downside with using a noisy network is its possible running time in eliminating external errors: the noisy neural network may need more iterations to achieve the same error correction performance. Interestingly, our empirical experiments show that in certain scenarios, even the running time improves when using a noisy network.

Thm. 2 indicates that noisy neural networks (under our model) outperform noiseless ones, but does not specify the level of errors that such networks can correct. Now we derive a theoretical upper bound on error correction performance. To this end, let $P_{c_i}$ be the average probability that a cluster can correct $i$ external errors in its domain. The following theorem gives a simple condition under which Alg. 2 can correct a linear fraction of external errors (in terms of $n$) with high probability. The condition involves $\tilde{\lambda}$ and $\tilde{\rho}$, the degree distributions of the contracted graph $\tilde{G}$.

**Theorem 3.** *Under the assumptions that graph $\widetilde{G}$ grows large and it is chosen randomly with degree distributions given by $\widetilde{\lambda}$ and $\widetilde{\rho}$, Alg. 2 is successful if*

$$\epsilon\widetilde{\lambda}\left(1 - \sum_{i \geq 1} P_{c_i}\frac{z^{i-1}}{i!} \cdot \frac{d^{i-1}\widetilde{\rho}(1-z)}{dz^{i-1}}\right) < z, \ for \ z \in [0, \epsilon]. \tag{4}$$

Proof is given in [23] and is based on the density evolution technique [25]. Thm. 3 states that for any fraction of errors $\Lambda_{\upsilon,\nu} \leq \Lambda_{\upsilon,\nu}^*$ that satisfies the above recursive formula, Alg. 2 will be successful with probability close to one. Note that the first fixed point of the above recursive equation dictates the maximum fraction of errors $\Lambda_{\upsilon,\nu}^*$ that our model can correct. For the special case of $P_{c_1} = 1$ and $P_{c_i} = 0, \forall i > 1$, we obtain $\epsilon\widetilde{\lambda}1 - \widetilde{\rho}(1-z)) < z$, the same condition given in [9]. Thm. 3 takes into account the contribution of all $P_{c_i}$ terms and as we will see, their values change as we incorporate the effect of internal noise $\upsilon$ and $\nu$. Our results show that the maximum value of $P_{c_i}$ does not occur when the internal noise is equal to zero, i.e. $\upsilon = \nu = 0$, but instead when the neurons are contaminated with internal noise! As an example, Fig. 2 illustrates how $P_{c_i}$ behaves as a function of $\upsilon$ in the network considered (note that maximum values are not at $\upsilon = 0$). This finding suggests that even individual clusters are able to correct more errors in the presence of internal noise.

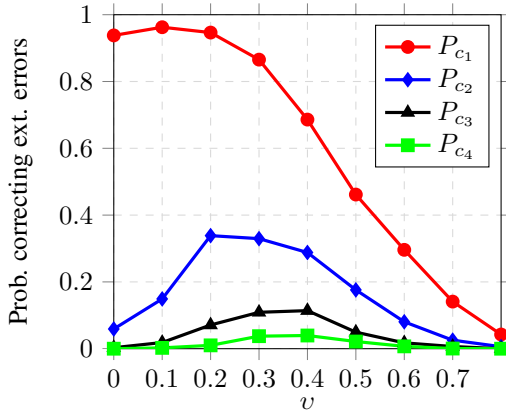
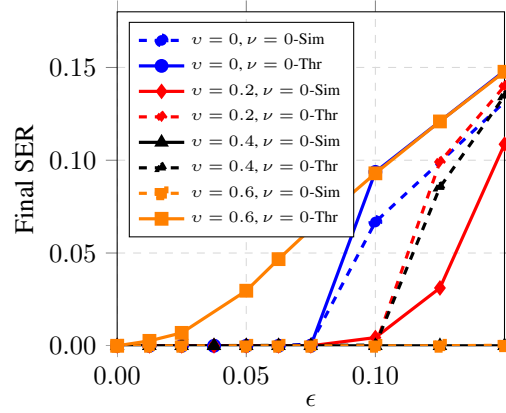

Figure 2: The value of $P_{c_i}$ as a function of pattern neurons noise $\upsilon$ for $i = 1, \ldots, 4$. Noise at constraint neurons is assumed as zero ($\nu = 0$).

Figure 3: The final SER for a network with $n = 400$, $L = 50$ cf. [9]. The blue curves correspond to the noiseless neural network.

## 3.1 Simulations

Now we consider simulation results for a finite system. To learn the subspace constraints (1) for each cluster $G^{(\ell)}$ we use the learning algorithm in [9]. Henceforth, we assume that the weight matrix $W$ is known and given. In our setup, we consider a network of size $n = 400$ with $L = 50$ clusters. We have 40 pattern nodes and 20 constraint nodes in each cluster, on average. External error is modeled by randomly generated vectors $z$ with entries $\pm 1$ with probability $\epsilon$ and 0 otherwise. Vector $z$ is added to the correct patterns, which satisfy (1). For recall, Alg. 2 is used and results are reported in terms of Symbol Error Rate (SER) as the level of external error ($\epsilon$) or internal noise ($\upsilon, \nu$) is changed; this involves counting positions where the output of Alg. 2 differs from the correct pattern.

### 3.1.1 Symbol Error Rate as a function of Internal Noise

Fig. 3 illustrates the final SER of our algorithm for different values of $\upsilon$ and $\nu$. Recall that $\upsilon$ and $\nu$ quantify the level of noise in pattern and constraint neurons, respectively. Dashed lines in Fig. 3 are simulation results whereas solid lines are theoretical upper bounds provided in this paper. As evident, there is a threshold phenomenon such that SER is negligible for $\epsilon \leq \epsilon^*$ and grows beyond this threshold. As expected, simulation results are better than the theoretical bounds. In particular, the gap is relatively large as $\upsilon$ moves towards one.

A more interesting trend in Fig. 3 is the fact that internal noise helps in achieving better performance, as predicted by theoretical analysis (Thm. 2). Notice how $\epsilon^*$ moves towards one as $\nu$ increases.

This phenomenon is examined more closely in Figs. 4a and 4b where $\epsilon$ is fixed to $0.125$ while $\upsilon$ and $\nu$ vary. As we see, a moderate amount of internal noise at both pattern and constraint neurons improves performance. There is an optimum point $(\upsilon^*, \nu^*)$ for which the SER reaches its minimum. Fig. 4b indicates for instance that $\nu^* \approx 0.25$, beyond which SER deteriorates.

## 3.2 Recall Time as a function of Internal Noise

Fig. 5 illustrates the number of iterations performed by Alg. 2 for correcting the external errors when $\epsilon$ is fixed to $0.075$. We stop whenever the algorithm corrects all external errors or declare a recall error if all errors were not corrected in 40 iterations. Thus, the corresponding areas in the figure where the number of iterations reaches 40 indicates decoding failure. Figs. 6a and 6b are projected versions of Fig. 5 and show the average number of iterations as a function of $\upsilon$ and $\nu$, respectively.

The amount of internal noise drastically affects the speed of Alg. 2. First, from Fig. 5 and 6b observe that running time is more sensitive to noise at constraint neurons than pattern neurons and that the algorithms become slower as noise at constraint neurons is increased. In contrast, note that internal noise at the pattern neurons may improve the running time, as seen in Fig. 6a.

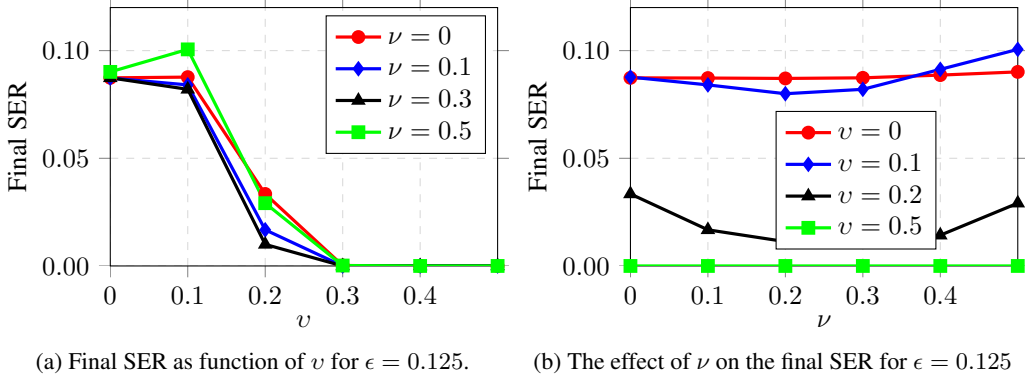

(a) Final SER as function of $\upsilon$ for $\epsilon = 0.125$.

(b) The effect of $\nu$ on the final SER for $\epsilon = 0.125$

Figure 4: The final SER vs. internal noise parameters at pattern and constraint neurons for $\epsilon = 0.125$

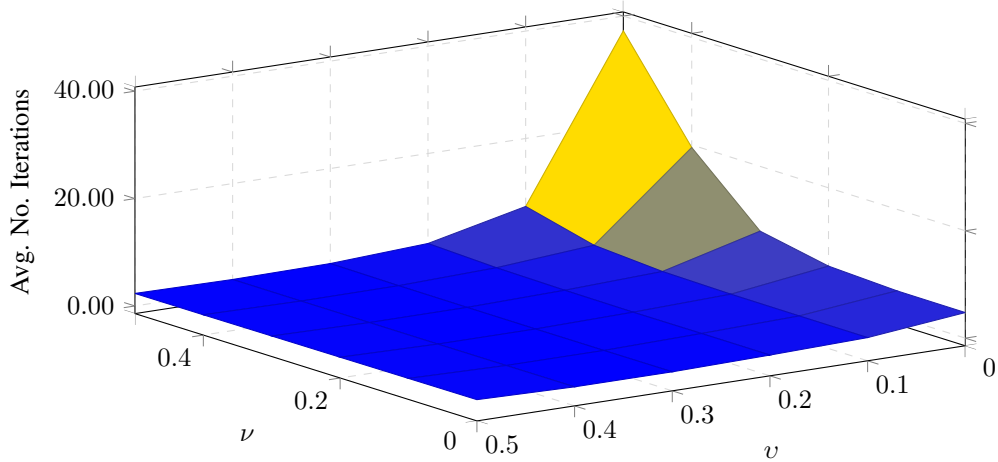

Figure 5: The effect of internal noise on the number of iterations of Alg. 2 when $\epsilon = 0.075$.

Note that the results presented here are for the case where the noiseless decoder succeeds as well and its average number of iterations is pretty close to the optimal value (see Fig. 5). In [23], we provide additional results corresponding to $\epsilon = 0.125$, where the noiseless decoder encounters stopping sets while the noisy decoder is still capable of correcting external errors; there we see that the optimal running time occurs when the neurons have a fair amount of internal noise.

In [23] we also provide results of a study for a slightly modified scenario where there is only internal noise and no external errors. Furthermore, $\varphi < \upsilon$. Thus, the internal noise can now cause neurons to make wrong decisions, even in the absence of external errors. There, we witness the more familiar phenomenon where increasing the amount of internal noise results in a worse performance. This finding emphasizes the importance of choosing update threshold $\varphi$ and $\psi$ according to Lem. 1.

## 4 Pattern Retrieval Capacity

For completeness, we review pattern retrieval capacity results from [9] to show that the proposed model is capable of memorizing an exponentially large number of patterns. First, note that since the patterns form a subspace, the number of patterns $C$ does not have any effect on the learning or recall algorithms (except for its obvious influence on the learning time). Thus, in order to show that the pattern retrieval capacity is exponential in $n$, all we need to demonstrate is that there exists a training set $\mathcal{X}$ with $C$ patterns of length $n$ for which $C \propto a^{rn}$, for some $a > 1$ and $0 < r$.

**Theorem 4** ( [9]). *Let $\mathcal{X}$ be a $C \times n$ matrix, formed by $C$ vectors of length $n$ with entries from the set $\mathcal{S}$. Furthermore, let $k = rn$ for some $0 < r < 1$. Then, there exists a set of vectors for which $C = a^{rn}$, with $a > 1$, and $rank(\mathcal{X}) = k < n$.*

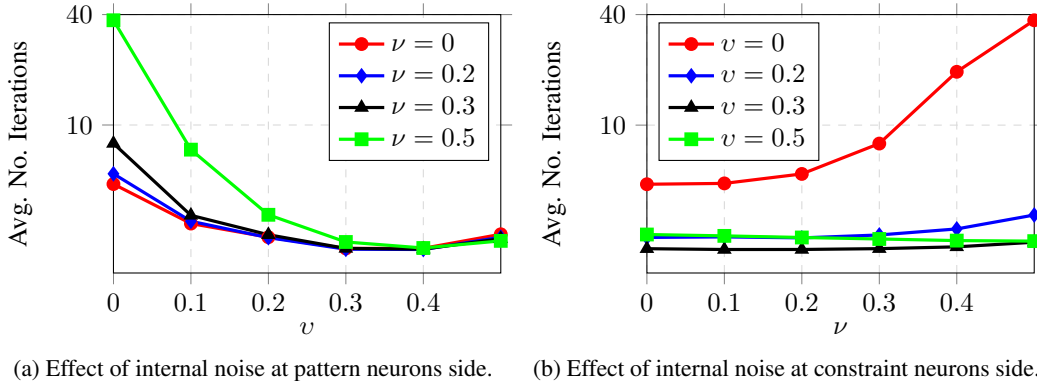

(a) Effect of internal noise at pattern neurons side.     (b) Effect of internal noise at constraint neurons side.

Figure 6: The effect of internal noise on the number of iterations performed by Alg. 2, for different values of $\upsilon$ and $\nu$ with $\epsilon = 0.075$. The average iteration number of $40$ indicate the failure of Alg. 2.

The proof is constructive: we create a dataset $\mathcal{X}$ such that it can be memorized by the proposed neural network and satisfies the required properties, i.e. the subpatterns form a subspace and pattern entries are integer values from the set $\mathcal{S} = \{0, \dots, S-1\}$. The complete proof can be found in [9].

## 5 Discussion

We have demonstrated that associative memories with exponential capacity still work reliably even when built from unreliable hardware, addressing a major problem in fault-tolerant computing and further arguing for the viability of associative memory models for the (noisy) mammalian brain. After all, brain regions modeled as associative memories, such as the hippocampus and the olfactory cortex, certainly do display internal noise [12, 13, 26]. The linear-nonlinear computations of Alg. 1 are certainly biologically plausible, but implementing the state reversion computation of Alg. 2 in a biologically plausible way remains an open question.

We found a threshold phenomenon for reliable operation, which manifests the tradeoff between the amount of internal noise and the amount of external noise that the system can handle. In fact, we showed that internal noise actually improves the performance of the network in dealing with external errors, up to some optimal value. This is a manifestation of the *stochastic facilitation* [13] or *noise enhancement* [14] phenomenon that has been observed in other neuronal and signal processing systems, providing a functional benefit to variability in the operation of neural systems.

The associative memory design developed herein uses thresholding operations in the message-passing algorithm for recall; as part of our investigation, we optimized these neural firing thresholds based on the statistics of the internal noise. As noted by Sarpeshkar in describing the properties of analog and digital computing circuits, "In a cascade of analog stages, noise starts to accumulate. Thus, complex systems with many stages are difficult to build. [In digital systems] Round-off error does not accumulate significantly for many computations. Thus, complex systems with many stages are easy to build" [27]. One key to our result is capturing this benefit of digital processing (thresholding to prevent the build up of errors due to internal noise) as well as a modular architecture which allows us to correct a linear number of external errors (in terms of the pattern length).

This paper focused on recall, however learning is the other critical stage of associative memory operation. Indeed, information storage in nervous systems is said to be subject to storage (or learning) noise, *in situ* noise, and retrieval (or recall) noise [11, Fig. 1]. It should be noted, however, there is no essential loss by combining learning noise and *in situ* noise into what we have called external error herein, cf. [19, Fn. 1 and Prop. 1]. Thus our basic qualitative result extends to the setting where the learning and stored phases are also performed with noisy hardware.

Going forward, it is of interest to investigate other neural information processing models that explicitly incorporate internal noise and see whether they provide insight into observed empirical phenomena. As an example, we might be able to understand the threshold phenomenon observed in the SER of human telegraph operators under heat stress [28, Fig. 2], by invoking a thermal internal noise explanation.

## Footnotes

[1] Note that the proposed algorithms also work with larger noise values, i.e. from a set $\{-a, \dots, a\}$ for some $a \in \mathbb{N}$, see [23]; the $\pm 1$ noise model is presented here for simplicity.

[2]Note that $x_i^{(\ell)}(t+1)$ is further mapped to the interval $[0, S-1]$ by saturating the values below 0 and above $S - 1$ to 0 and $S - 1$ respectively. The corresponding equations are omitted for brevity.

[3]Note that although the values of $y_i^{(\ell)}(t)$ can be shifted to $0, 1, 2$, instead of $-1, 0, 1$ to match our assumption that neural states are non-negative, we leave them as such to simplify later analysis.

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
