[Reviews · NeurIPS 2013]

Submitted by Assigned_Reviewer_3

Classic work on associative memories following Hopfield 1982 focused on issues of capacity and performance, usually considering random memories embedded as stable attractors of a dynamical system. Such work usually led to capacities which scale linearly with the size of the network. The present work proposes a neural architecture which is able to reach exponential capacities at the cost of introducing specific low-dimensional structure into the stored patterns.
The authors propose a bi-partite architecture of pattern and constraint neurons corresponding to patterns and clusters, and a two-tiered algorithm, based on within and between-module processing aimed at retrieving clean versions of noise corrupted the patterns. The intra-module algorithm operates iteratively based on forward and backward iterations, based on a belief variable. The inter-module algorithm is based on iterations taking place until all constraints are satisfied. The authors present theoretical results relating to convergence, showing that there is a noise range for the internal dynamics, which leads to potentially improved performance relative to noise-free dynamics. Simulation results seem to corroborate these theoretical claims, also demonstrating an interesting threshold effect as a function of the noise parameter.
The results presented here are interesting and, to the best of my knowledge, novel. The fact that noise improves performance in such a nonlinear dynamical system is indeed surprising and non-trivial. While the authors have tried to present some intuition about their results (the proof of which appears in the appendix), I was not able to get a good feeling for ‘what makes the system tick’.
Specific comments: The authors do not seem to be aware that a great deal of early work on associative memories dealt with stochastic retrieval dynamics, as well as in structured patterns leading to higher capacities. AN early review of many of these results can be found in Daniel Amit’s “Modeling Brain Function”, and later work. In fact, the advantageous effect of noisy dynamics on improving retrieval by eliminating so-called spurious states was noted there.
Although the authors have shown that noise may assist in improving retrieval, it would be nice to understand the robustness of this result. For example, what would happen if the weights W_{ij} were corrupted by noise? Such robustness had been shown in early neural network studies (again, see Amit’s book).
Finally, the dynamics given by eq. (2) and (3) seem to be at odds with the requirement that the patterns be restricted to a fixed finite range {0,…,S-1}. How is this guaranteed by this update rule?
Summary: A work which suggests how to increase the capacity of associative memories exponentially at the price of introducing specific structure into the stored patters. The authors do not relate to earlier work on the effect of noise, but present an interesting result on possible merits of noise.

Submitted by Assigned_Reviewer_6

An associative memory mechanism for pattern completion of non-binary integer valued data is proposed for which the authors claim that the number of stored patterns can be exponential in the number of neurons. The main result of this paper is an analysis of the performance of this memory model in the presence of intrinsic neural noise. Interestingly it is shown by the analysis and simulations that intrinsic noise can increase the performance of the model.

Comments:

1) It would be important to add results supporting that the model can store exponentially many patterns. Either theoretical capacity results or simulation curves how recall error behaves as a function of the number of stored patterns.

2) To make the paper stand-alone, it would be important to describe the learning algorithm from [10] at least the main principle

3) For integer-valued data it is somewhat artificial to confine the (external) additive noise to be -1, 0 or 1. Can the model handle larger additive errors?

4) The discussion stresses the neurobiological relevance of this model. You also should discuss that many elements of your algorithm are not biologically plausible, e.g., the negative activity in the control neurons, the algorithm loops Alg1 and Alg 2 etc.

5) It would be important in the final version of the paper to include the essential proof lines from the appendix.

6) To gain space, you could omit the 3D figure, Fig 4, which is redundant with Fig. 5.


Summary: An associative memory mechanism for pattern completion of non-binary integer valued data is proposed for which the authors claim that the number of stored patterns can be exponential in the number of neurons.

Submitted by Assigned_Reviewer_7

SUMMARY

In this article, the authors study a model for associative memory where the patterns lie in some low-dimensional manifold. The structure of the patterns is captured by the so-called "constraint-neurons". Those constraint-neurons together with the pattern neurons form the memory network. The authors propose a recall algorithm for this network and show that intrinsic noise actually improves recall performance.

The paper is well written and as far as I can judge, it is technically sound.

MAJOR COMMENTS

1. Biological plausibility. The paper starts with considerations on associative memory and the brain. However, the associative memory model seems very distant to any biological implementation.
- For example, how would step 4 in Algorithm 2 be implemented in some biologically realistic network? This step assumes that some memory should be kept over the tmax iterations of Algo 1 (c.f. Step 3 in Algo 3).
- How should we think of the integer-valued state of the neuron (are states spike vs non-spike? this binary code (S = 2) would be different from the code in the constraint neurons.)


2. As mentioned by the authors, the architecture is similar to the one of Restricted Boltzman machines (RBM), the pattern neurons being the visible neurons and the constraint neuron being the hidden neurons. Now, with RBM, neurons are also intrinsically noisy and the level of intrinsic noise is adapted (through synaptic strength) such that the marginal distribution over visible units is consistent with the data. So I would also expect that the performance would be poorer with RBM if we forced neurons to be deterministic. Therefore, it is not clear to what extend it is a breaking news that intrinsic noise is good. Furthermore, the authors state that Deep Belief Nets (and therefore RBM as well) are used for classification and not pattern completion, but this is not correct. RBM can be used for pattern completion.


3 Other approaches to associative memories have been proposed. For example in [R1], Lengyel et al. see recall as a Bayesian inference process which exploits knowledge about the structure of the stored patterns (prior) and the knowledge about the learning rule as well as knowledge about the corrupting process of the inputs (the likelihood). In this perspective, the posterior distribution is a deterministic function of the prior and the likelihood and does not require additional noise. So how is it possible that the "best" recall rule (in a Bayesian sense) does not need noise (except if one want to sample from the posterior distribution) and the recall rule presented here needs noise?

[R1] Lengyel, M., Kwag, J., Paulsen, O., & Dayan, P. (2005). Matching storage and recall: hippocampal spike timing–dependent plasticity and phase response curves. Nature Neuroscience, 8(12), 1677–1683.

Summary: The paper is well written and as far as I can judge, it is technically sound.
Author Feedback

Author rebuttal: We kindly thank the anonymous reviewers for their careful review of our work. Our responses to the issues raised can be found below.

Reviewer 3
We regret overlooking classical contributions described in Amit's book which are related to our efforts and have added a reference. We have also provided a comparison between our work and previous art. We will discuss this more thoroughly in an extended version of the paper.

Regarding structured patterns, we plan to discuss works of Jankowski (1996) and Muezzinoglu (2003) on multistate complex-valued neural networks and their capacities in an extended version of the paper. Here we mention a recent line of work that aims to exploit inherent pattern structure either by using correlations among patterns or by memorizing only patterns with some sort of redundancy. Pioneering this approach, Gripon (2011) demonstrated considerable improvements in pattern retrieval barriers of Hopfield networks, albeit still not passing polynomial capacity.

To the best of our knowledge, our work is the first to rigorously show not only that robust and exponential capacity is possible, but even more surprisingly a certain amount of internal noise (which we again analytically characterized) can even help the neural network. This type of rigorous analysis for neural networks is absent in many of the previous work. In particular, in Amit's book and many follow-ups, the focus is on retrieving random patterns whereas in our work the focus is on correlated patterns (this is why we obtain an exponential capacity). Moreover, the effect of internal noise was previously studied through numerical simulations whereas we derive the exact analytical conditions.

Studying robustness against weights W_{ij} is an interesting future direction we hope to pursue. One possible approach is to include the net corruption due to weight discrepancies in the total internal noise of a neuron, as they both affect the input sum received by each neuron.

The reviewer had raised the issue that pattern and constraint update rules seem at odds with the requirement that patterns be restricted to a fixed finite range {0,…,S-1}. For equation (2), we must truncate values below 0 and above S-1 to 0 and S-1, respectively. For equation (3), the values -1,0,1 are chosen to facilitate presenting Alg. 1. Otherwise, one can use 0,1,2 and simply downshift each y_i^{(\ell)} by 1 to obtain -1,0,1 values in Eq (3).

Reviewer 6
The reviewer had asked us to include results supporting the fact that the model can store exponentially many patterns. Theorem 4 in Section 4 has been included to show exactly this.

The reviewer asked us to describe the learning algorithm for our system. This has been sketched in Section 2.

Regarding whether the model handles larger additive errors: The simple -1,+1 noise model was used to illustrate the principles of the recall phase. Our analysis can be easily extended to larger additive noise. We have conducted a new set of experiments with larger additive noise which will be included in an extended version of the paper.

Regarding biological plausibility of our algorithm, we acknowledge certain operations in the sequential peeling algorithm are not too plausible for neural systems. Notwithstanding, note that neural implementations of belief propagation and similar message-passing decoding algorithms have been suggested previously (e.g., Beck et al, 2007; Dayan et al, 1995; Deneve, 2008; Doya et al, 2007; Hinton et al, 1986; Ma et al, 2006; Litvak et al, 2009). We have included a brief discussion of this point in the paper.

Reviewer 7
Further regarding biological plausibility, Step 4 in Alg. 2 is indeed the hardest part to map to a biological neural network, but please see (Druckmann et al, 2012) on stable representations which suggests that keeping a state for a short amount of time is realistic for some neurons.

The integer-valued state of the neuron can be thought of as the short-term firing rate of the neuron. The same model can be easily modified to account for spike timings as well.

We agree with the reviewer that Restricted Boltzmann Machines (RBMs) are also robust against noise in several practical situations. However, to the best of our knowledge, no rigorous proof has ever been provided. In fact, our proof technique might lead to analytical analysis of RBMs in the face of external and internal noise.

The reviewer asks why the "best" Bayesian recall rule does not need noise but the recall rule here needs noise. A minor correction is that we do not claim the recall phase requires noise; we state that under the model considered herein (bipartite, multi-level, etc), with the proposed iterative denoising algorithm, not only do we gain exponential capacity and robustness, but also internal noise helps. We analytically find the threshold below which internal noise is helpful and above which it is harmful. Please note that optimal (best) Bayesian inference in non-tree networks is provably hard and so we must opt for simpler strategies like iterative decoding proposed herein or an approximation to the MAP inference problem as in (Lengyel et al, 2005). As such, these algorithms only guarantee to find a local maximum and hence introducing internal noise usually helps to avoid some of the local maxima.